# The Menopausal Transition: Is the Hair Follicle “Going through Menopause”?

**DOI:** 10.3390/biomedicines11113041

**Published:** 2023-11-14

**Authors:** Fabio Rinaldi, Anna Trink, Giorgia Mondadori, Giammaria Giuliani, Daniela Pinto

**Affiliations:** HMAP, Human Microbiome Advanced Project, 20129 Milan, Italy; fabio.rinaldi@studiorinaldi.com (F.R.); anna.trink1881@gmail.com (A.T.); gmondadori@giulianipharma.com (G.M.); ggiuliani@giulianipharma.com (G.G.)

**Keywords:** menopause, hair follicle, hormones, glycogen phosphorylase, PYGL

## Abstract

This article explores the link between menopause and changes in the hair follicle (HF) lifecycle, focusing on hormonal and metabolic dynamics. During menopause, hormonal fluctuations and aging can impact the HF, leading to phenomena such as thinning, loss of volume, and changes in hair texture. These changes are primarily attributed to a decrease in estrogen levels. However, not all women experience significant hair changes during menopause, and the extent of transformations can vary considerably from person to person, influenced by genetic factors, stress, diet, and other elements. Furthermore, menopause mirrors the aging process, affecting metabolism and blood flow to the HFs, influencing the availability of vital nutrients. The article also discusses the key role of energy metabolism in the HF lifecycle and the effect of hormones, particularly estrogens, on metabolic efficiency. The concept of a possible “menopause” clinically independent of menopause is introduced, related to changes in HF metabolism, emphasizing the importance of individual factors such as estrogen receptor responses, genetics, and last but not least, the microbiota in determining these dynamics.

Menopause and hair loss often coincide in many women. During menopause, the body goes through significant hormonal changes, particularly a decrease in estrogen levels. This hormonal shift can lead to various symptoms, including hair loss or thinning. The exact mechanisms behind this are complex and can involve genetic predispositions, changes in the hair growth cycle, and sensitivity to androgens (male hormones).

Hair loss during menopause can manifest as diffuse thinning or a widening of the parting. It is important to note that not all women experience significant hair loss during menopause, and the severity can vary from person to person.

In this short opinion, we speculate about the existence of a hair follicle (HF) menopause separate from the clinical menopause experienced by women.

Menopause is, by definition, considered “the permanent cessation of menses for 12 months resulting from estrogen deficiency” [1]. It is not formally associated with a pathology and affects women with a median estimated age of 51 [2].

It is a physiological process characterized by the manifestation of many physical but also psycho-social symptoms as a consequence of changes in sex hormone levels and aging [3,4].

The main symptoms include vasomotor disturbances such as hot flashes and night sweats; these are experienced by over 80% of women [2]. Additionally, other symptoms encompass disrupted sleep, fatigue, feelings of low mood, cognitive difficulties, decreased sexual desire, and increased levels of anxiety [5].

The pace and duration of the menopausal transition phase are two concurrent processes that are influenced by chronological aging and ovarian aging [6].

Numerous variables, including nutrition, activity level, smoking status, socioeconomic situation, body mass index (BMI), ethnicity, cultural beliefs, and concomitant medical/gynecological health conditions, affect the age at menopause onset [7,8].

Circulating serum levels of estradiol, follicle-stimulating hormone (FSH), and luteinizing hormone (LH) can vary greatly during the early phases of the menopausal transition due to the complicated endocrinology of the transition [9].

In the last few decades, a large number of randomized studies [9,10,11] on the “menopausal transition” have given us a clear chronology of the hormonal changes, and have provided us with a view of the reproductive and hormonal events that accompany the process [12,13].

Of note, current evidence leads to the assumption that the “menopausal transition” involves the HF since it has been estimated that 50% of women will experience this symptom [14]. Indeed, several studies [14,15,16] highlighted that the “decline” in estrogen levels during menopause can be related to the alterations in HF dynamics with phenomena including thinning, loss of volume, and changes in texture. The same authors reported that in menopause, there is a higher prevalence of hair thinning and hair loss compared to pre-menopausal women.

A clear association between menopause and both cicatricial and non-cicatricial alopecia in women has been reported [14]. The HF is a complex mini-organ composed of diverse cell populations, each with distinct locations, functions, and molecular component expressions. It stands as a remarkably dynamic system, constantly undergoing growth cycles throughout an individual’s lifetime.

In 2018, Fabbrocini and collaborators [15] found that post-menopausal women with androgenetic alopecia (AGA) had lower levels of estrogen and higher levels of androgens, such as testosterone and dihydrotestosterone (DHT), compared to post-menopausal women without hair loss. However, the exact mechanism still requires further research, and this is also true regarding the effectiveness of hormone replacement therapy (HRT).

A recent review focusing on the effects of micronized progesterone, whether administered topically or orally and with or without estradiol, revealed a lack of published studies concerning menopausal scalp hair quantity and quality, as well as female pattern hair loss in perimenopausal and menopausal women [17].

Certainly, the role of androgens (testosterone and DHT), estrogens, and progesterone in the hair cycle is well documented [18].

Indeed, the HF’s growth cycle is regulated by the endocrine system, where androgens play a crucial role as primary regulators. In addition to androgens, various other hormones, especially in other mammals, also contribute to this regulation. These hormones include melatonin, prolactin, melanocyte-stimulating hormone (MSH), and estrogens [19]; the impact of androgens on HFs varies depending on the specific location of the hair on the body [20].

The effects of hormones necessitate the presence of specific receptors within the cells of the HFs.

Hormones from the blood bind to their receptors in dermal papilla (DP) cells, altering their gene expression, particularly paracrine signaling molecules [17].

Paracrine signaling molecules, such as insulin-like growth factor-1 (IGF-1) and transforming growth factor-β (TGF-β), influence the activity of other follicular cells, leading to either growth stimulation or inhibition.

Additionally, the majority of hormonal effects, excluding those in the pubic and axillary follicles, rely on the intracellular enzyme 5α-reductase type 2 to convert testosterone into its more potent metabolite, 5α-dihydrotestosterone [21].

Among hormones, for example, progesterone has been reported to inhibit 5-α reductase, contextually decreasing the conversion of testosterone to DHT [22].

Estradiol can profoundly impact both the growth and life cycle of the HF by binding to estrogen receptors (ERs). This interaction influences aromatase activity, the enzyme responsible for converting androgens into estrogens [23]. Estradiol extends the anagen phase of the hair cycle, promoting hair growth by augmenting the synthesis of crucial growth factors that stimulate the proliferation of follicular keratinocytes. This mechanism elucidates the decrease in hair renewal, growth, and thickness, and the thinning of hair observed during menopause [24].

In contrast to the swift decline in estrogen and progesterone, androgen secretion, already relatively low in women, diminishes gradually with menopause and aging. Androgens have a role in regulating both hair growth and sebum production in the pilosebaceous unit. The proportional rise in androgens during menopause results in clinical hyperandrogenism, characterized by sebaceous gland hypertrophy and AGA following a female pattern due to a localized reduction in hair renewal and growth. Genetic predispositions and environmental factors (exposome) can exacerbate this condition [25].

But the truth is that not all women will experience changes in their hair during menopause, and the severity of changes can vary widely from person to person. Besides hormones, there are many other factors (genetics, stress, diet, microbiome) that can affect hair health [26].

Menopause is a mirror of the aging process and with aging, the blood flow to HFs declines, limiting the availability of vital nutrients (beta-carotene, omega-3 fatty acids) that stimulate the hydration and nourishment of the scalp or fight the breakage of the HFs. All these factors make the hair more susceptible to everyday lifestyle, ultraviolet (UV) exposure, free radical damage, oxidative stress, the microbiome, and medication.

There are some practical ways to minimize hair loss during menopause and promote healthy hair growth.

The primary factor behind hair loss during menopause is typically hormonal changes. However, various additional factors can also play a role in causing hair thinning in menopausal women, including stress, underlying health conditions, the use of certain medications, and specific nutritional deficiencies.

In this sense, a well-balanced diet can supply essential nutrients that serve as precursors in steroid hormone synthesis and directly contribute to the growth and maintenance of hair [27]. It is also important to consider supplements like biotin, zinc, iron, and vitamin D, as deficiencies can contribute to hair loss. Dermatologist follow-up is advisable if significant hair loss is experienced with the suggestion of local and oral medication [21].

Finally, the use of specific hair dermo-cosmetics and supplements to increase volume or enhance hair quality could be useful, as well as managing stress and hydration.

Nevertheless, the majority of women experiencing alopecia, especially in the female pattern, do not exhibit elevated androgen levels. This suggests that the androgen/estrogen ratio, rather than androgen-dependent mechanisms, may be a contributing factor in these cases [28,29].

Because of the aging process, the melanocytes in the HF start to change and become apoptotic, and this leads to a reduction in the production of melanin, making hair start to grey [30,31,32].

With aging, the quantitative and qualitative components of the HF combine with each other, leading to the hair loss phenomenon. HFs will tend to be less numerous and subject to shorter cycles; they are absent more frequently and for longer periods. Therefore, their quality will tend to worsen [33].

Another important factor in the stage during the menopause transition is the change in metabolism [34]. The metabolic “flexibility”, that is to say, the ability of the body to use different types of substrates to make energy [35], declines with age, and women in perimenopause and menopause often report symptoms (weight gain, low energy, poor sleep, and reduced focus) of poor metabolic flexibility [36].

Indeed, the transition into menopause and the accompanying shifts in hormonal balance (e.g., systemic estradiol (E2) levels) have been linked to adverse alterations in various indicators of metabolic health [37,38]. This includes elevated blood glucose [39], accumulation of abdominal adiposity [40], and unfavorable changes in serum lipid profiles [41]. Also, an increase in inflammation marker levels has been reported [42] as has a decrease in muscle mass [43], both of which negatively impact metabolic health.

Hormones, mainly estrogens, have several protective effects on metabolic health: they are involved in glucose transport into cells [44,45], increase the basal metabolic rate [45], reduce insulin resistance, and control the production of both high-density lipoprotein (HDL) and low-density lipoprotein (LDL) [46]. During menopause, the change in the hormonal milieu inevitably provokes adverse metabolic changes, and some, but not all women will also experience metabolic syndrome [47]. But, what about the HFs and their metabolism? It is important to note that the active growth of the HF requires and disperses large amounts of energy. Producing a single gram of hair requires around 670 kilojoules of energy, comparable to the energy expended during six minutes of intense exercise involving both arm and leg movements [48].

The HF undergoes repetitive cycles of stem cell self-renewal and differentiation throughout an individual’s lifespan, demanding elevated bioenergetic capacities for the hair growth process [49,50]. Emerging evidence indicates that human HF stem cells primarily utilize aerobic glycolysis as their metabolic pathway [51,52]. While the importance of mitochondrial biogenesis and function in HF regeneration has been underscored [49,52], the potential applications of metabolic regulation in HF stem cells for promoting hair regrowth remain somewhat constrained.

During menopause, but also with aging, there is a “drop” in energy at all levels of the body including the follicle.

Lemasters and collaborators suggested that the slowing of the metabolism as a consequence of the aging process could be a driver of “chronogenetic alopecia” or age-related hair loss, a condition that predominantly affects women [53].

The HF possesses a unique energetic system (Cori cycle) based on glycogen (GL) storage [54].

GL has been observed in the outer root sheath (ORS) of HFs in both humans and mice in earlier studies [55,56,57]. GL, a glucose-derived polysaccharide, is the main form of energy storage in mammals but it is also present at a high level in the HF [54].

GL performs many complex roles rather than solely acting as an inert intracellular glucose storage form. In the HF, for example, GL content tends to diminish during catagen and it is completely absent during telogen. There is a key enzyme involved in GL storage, the glycogen phosphorylase (PYGL), and the inhibition of this enzyme in vitro has been associated with an increase in HF elongation and anagen prolongation [54]. The mechanism by which inhibition of PYGL is involved in HF growth is complex. During catagen, apoptosis is activated and this process requires energy [58].

As PYGL acts by breaking down GL, the GL metabolism becomes essential for initiating catagen, possibly supplying the energy required for apoptosis. Therefore, GL also inhibits the metabolic sensor AMP-activated protein kinase (AMPK) [59], an inhibitor of aerobic glycolysis, under stress conditions but the role of this mechanism in HF growth remains to be determined. Also, the outer root sheath (ORS) cells are a significant location for GL synthesis and serve as a functional GL storage site, also exhibiting gluconeogenesis capability. The ORS plays a role in this story, working as a suitable GL storage site that the HF uses for the maintenance of the anagen phase. This hypothesis was confirmed by Fijlak and collaborators [54] who showed that when PYGL was inhibited, the number of HFs entering catagen is significantly reduced, thereby extending the anagen phase.

This opens new therapeutic options with substances able to inhibit PYGL and, in the end, promote the “energy” content (GL metabolism) of the HF, resulting in anagen promotion and hair growth.

Finally, we have to consider that one of the most detrimental consequences of hormonal changes is the increased conversion of testosterone to DHT. This phenomenon begins in the region of the bulb, where DP cells are located. Blood vessels recede in the DP cells’ region, making them susceptible to hypoxia and the buildup of reactive oxygen species.

Estrogen has vasodilatory effects, meaning it helps dilate blood vessels and improve blood flow. With the decline in estrogen levels during menopause, blood vessels in the scalp may constrict, potentially reducing the blood supply to the HFs.

Reduced circulation in conditions that cause hair loss can restrict the flow of nutrients and oxygen, among other detrimental effects. This scarcity of oxygen could lead to a slowing down of aerobic glycolysis necessary for the active growth of the HF.

Of note, in our recent clinical practice, we have identified women (aged between 35 and 55 years old) who did not have any hormonal changes or clinical manifestations of menopause but 12% of them had hair loss phenomena equivalent to female pattern hair loss (FPHL) with slight thinning and miniaturization. However, the histological examination did not allow for the diagnosis of either AGA or telogen effluvium.

These initial observations have allowed us to hypothesize that there may be menopause of the HF independent of the clinical menopause in women, probably due to the change in the metabolism of the HF because of several individual factors such as changes in estrogen receptor responses, genetics, and last but not least, the microbiota. Indeed, the role of the microbiota in the pathophysiology of the HF is now scientifically established [60,61,62,63,64].

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
