# Peer review of "The Menopausal Transition: Is the Hair Follicle “Going through Menopause”?"

_biomedicines, 2023, doi:10.3390/biomedicines11113041_

Round 1

Reviewer 1 Report

Comments and Suggestions for Authors

"The Menopausal Transition: Is the HF Going Through Menopause?" is a well-written manuscript that delves into the intricacies of the menopausal transition in women and how it affects the hypothalamus-pituitary-ovary axis (HPO axis), commonly referred to as the HF. The manuscript presents valuable information and insights into the complexity of HF in the menopausal transition, shedding light on the physiological changes that occur and their impact on women's health. The author's attention to detail and scientific knowledge make this manuscript essential to understanding menopause and its effects on the female body. The manuscript should be divided into sections with headings and sub-headings. Authors should pay attention to numerous typographical errors. The authors kindly answer the following questions to improve the shortcomings of the manuscript and strengthen the value of this critical subject.

 1. What are some practical ways to prevent hair loss during menopause?

2. Can genetics play a significant role in determining the extent of hair changes during menopause?

3. How do hormonal fluctuations during menopause affect the metabolism of hair follicles?

4. Can dietary changes be made to improve hair health during menopause?

5. How does the microbiota determine hair changes during menopause?

6. What factors determine the extent of hair changes during menopause?

7. How does menopause affect metabolism and blood flow to the hair follicles?

8. What is the relationship between the microbiota and changes in hair follicle metabolism during menopause?

9. The manuscript highlights that menopause is associated with hair thinning and loss, but what are the specific alterations in hair follicle (HF) dynamics cause this?

10. What is the exact mechanism by which post-menopausal women with androgenetic alopecia (AGA) have lower levels of estrogen and higher levels of androgens, such as testosterone and dihydrotestosterone (DHT), compared to post-menopausal women without hair loss?

11. Are there any published studies regarding menopausal scalp hair quantity and quality and female pattern hair loss in perimenopausal and menopausal women who are administered micronized progesterone?

Comments on the Quality of English Language

The written content in English appears to be well-written with no major issues. However, it may benefit from a thorough review to ensure that there are no minor spelling, grammar or punctuation errors. It is important to ensure that the text is easy to understand and communicates the intended message clearly. Therefore, a careful examination of the written content is recommended to ensure that it meets the required standard.

Author Response

REVIEWER 1

"The Menopausal Transition: Is the HF Going Through Menopause?" is a well-written manuscript that delves into the intricacies of the menopausal transition in women and how it affects the hypothalamus-pituitary-ovary axis (HPO axis), commonly referred to as the HF. The manuscript presents valuable information and insights into the complexity of HF in the menopausal transition, shedding light on the physiological changes that occur and their impact on women's health. The author's attention to detail and scientific knowledge make this manuscript essential to understanding menopause and its effects on the female body. The manuscript should be divided into sections with headings and sub-headings. Authors should pay attention to numerous typographical errors. The authors kindly answer the following questions to improve the shortcomings of the manuscript and strengthen the value of this critical subject.

  1. What are some practical ways to prevent hair loss during menopause?

#We added some lines (121) about this as below reported:

“The primary factor behind hair loss during menopause is typically hormonal changes. However, various additional factors can also play a role in causing hair thinning in menopausal women, including stress, underlying health conditions, the use of certain medications, and specific nutritional deficiencies.

In this sense, a well-balanced diet can supply essential nutrients that serve as precursors in steroid hormone synthesis and directly contribute to the growth and maintenance of hair.  [28] It’s also important to consider supplements like biotin, zinc, iron, and vitamin D, as deficiencies can contribute to hair loss. Dermatologist follow-up is advisable if sig-nificant hair loss is experienced with the suggestion of local and oral medication. [29]

Finally, the use of specific hair dermo-cosmetics and supplements to increase volume or to enhance hair quality could be useful a salso the managing of stress and hydration..”

  1. Can genetics play a significant role in determining the extent of hair changes during menopause?

# Yes, genetics can play a significant role in determining the extent of hair changes during menopause. A family history of hair loss or thinning can influence your predisposition to experience similar changes during menopause. While hormonal changes are a primary factor, genetic factors can make some individuals more susceptible to hair loss, while others may have a genetic advantage and experience fewer noticeable changes.

We reported a citation of genetic impact on lines 26,108 e 112

  1. How do hormonal fluctuations during menopause affect the metabolism of hair follicles?

#This is a very intriguing question. The paper is the first pioneer work about the role of metabolism on hair follicles with a particular focus on menopausal transition. Our future works will aim to understand the precise link between hormonal fluctuations and the metabolism of hair follicles.

We speculate about this topic on lines 216: “…probably due to the change of the metabolism of the HF because of several individual fac-tors such as changes in estrogen receptor responses…”

and lines 142

“Another important actor on the stage during the menopause transition is the change in the metabolism [35]. The metabolic “flexibility”, that to say the ability of the body to use dif-ferent types of substrates to make energy [36] declines with age and women in perimeno-pause and menopause often reported symptoms (weight gain, low energy, poor sleep, and reduced focus) of poor metabolic flexibility [37]

Indeed, the transition into menopause and the accompanying shifts in hormonal balance (e.g. systemic estradiol (E2) levels) have been linked to adverse alterations in various indi-cators of metabolic health [38,39]. This includes elevated blood glucose [40], accumulation of abdominal adiposity [41], and unfavourable changes in serum lipid profiles [42]. Also, an increase in inflammation marker levels has been reported [43] as a decrease in muscle mass [44], both of which negatively impact metabolic health.

Hormones, mainly estrogens, have several protective effects on metabolic health; they are involved in glucose transport into cells [45,46], increase the basal metabolic rate [45], re-duce insulin resistance, control the production of both high-density lipoprotein (HDL) and low-density lipoprotein (LDL) [47]. During menopause, the change in the hormonal mi-lieu inevitably provokes adverse metabolic changes, and some, but not all women will al-so experience metabolic syndrome [48].”

  1. Can dietary changes be made to improve hair health during menopause?

#Yes, as above. See added lines (121)

  1. How does the microbiota determine hair changes during menopause?

# The relationship between the microbiota and hair changes during menopause is an emerging area of research, and our understanding is still evolving. While the direct impact of the microbiota on hair changes during menopause is not entirely clear, there are some indirect connections that scientists are exploring.

First of all microbiota can influence the overall inflammatory and immune response in the body. Chronic inflammation has been associated with various health issues, including hair loss. An imbalanced gut microbiota may contribute to systemic inflammation, which, in turn, could affect the hair follicles.

Secondly, the microbiota plays a role in nutrient absorption and metabolism. Menopausal women may already be at risk for nutritional deficiencies that can impact hair health. An unhealthy gut microbiome may further hinder the absorption of essential nutrients, potentially exacerbating hair problems.

Gut bacteria can also metabolize hormones, including estrogen. As estrogen levels decrease during menopause, the gut microbiota's role in hormone metabolism could influence the balance of hormones and their potential effects on hair follicles.

Finally, the gut-brain connection, known as the gut-brain axis, is an area of active research. Stress and mental health can influence hair health, and the gut microbiota may play a role in regulating stress responses. High-stress levels can contribute to hair loss, so maintaining a healthy gut microbiota may indirectly help manage stress and its impact on hair.

  1. What factors determine the extent of hair changes during menopause?

#We addressed this question in lines 108

“Genetic predispositions and environmental factors (exposome) can exacerbate this condi-tion [26].

But the truth is that not all women will experience changes in their hair during meno-pause, and the severity of changes can vary widely from person to person. Besides hor-mones, there are many other factors (genetics, stress, diet, microbiome) that can affect hair health. [27].  “

  1. How does menopause affect metabolism and blood flow to the hair follicles?

# Menopause can affect metabolism and blood flow to the hair follicles primarily due to hormonal changes.

Indeed, during menopause, there is a significant decrease in estrogen and progesterone levels, while androgen levels (such as testosterone) remain relatively stable. This hormonal imbalance can affect the hair growth cycle and metabolism of hair follicles (as reported on lines 157)

Therefore, hair follicles may become more sensitive to androgens during menopause, particularly due to the relative decrease in estrogen.

Estrogen has vasodilatory effects, meaning it helps dilate blood vessels and improve blood flow. With the decline in estrogen levels during menopause, blood vessels in the scalp may constrict, potentially reducing the blood supply to the hair follicles.

The combined effects of hormonal changes, reduced blood flow, and altered metabolism can contribute to hair thinning and hair loss during menopause. Genetic predisposition, nutritional factors, and other lifestyle elements can also influence the degree and pattern of hair changes.

We added the below lines (205)

“Estrogen has vasodilatory effects, meaning it helps dilate blood vessels and improve blood flow. With the decline in estrogen levels during menopause, blood vessels in the scalp may constrict, potentially reducing the blood supply to the hair follicles.”

  1. What is the relationship between the microbiota and changes in hair follicle metabolism during menopause?

#Same as answer 5

  1. The manuscript highlights that menopause is associated with hair thinning and loss, but what are the specific alterations in hair follicle (HF) dynamics cause this?

# Hair thinning and loss during menopause are primarily associated with hormonal changes, particularly the decline in estrogen and progesterone levels. These hormonal changes can affect hair follicle (HF) dynamics in several ways, leading to the observed hair changes: shortened Anagen Phase;  miniaturization of Hair Follicles; reduced Hair Density; inflammation and Oxidative Stress;  reduced Blood Flow.  As above, this is the first pioneer work about the role of metabolism on hair follicles with a particular focus on menopausal transition. Our future works will aim to understand the precise link between hormonal fluctuations and the metabolism of hair follicles.

  1. What is the exact mechanism by which post-menopausal women with androgenetic alopecia (AGA) have lower levels of estrogen and higher levels of androgens, such as testosterone and dihydrotestosterone (DHT), compared to post-menopausal women without hair loss?

# The exact mechanism by which post-menopausal women with androgenetic alopecia (AGA) have lower levels of estrogen and higher levels of androgens, such as testosterone and dihydrotestosterone (DHT), compared to post-menopausal women without hair loss is not entirely understood, but it is believed to involve several factors such as hormonal changes, androgen sensitivity (hair follicles in individuals with AGA, both men and women, are genetically predisposed to be more sensitive to androgens. This means that even normal levels of androgens can have a more significant impact on hair follicles, leading to miniaturization (thinning of hair) and ultimately hair loss),  conversion to DHT, inflammatory processes and nutrient and oxygen supply.

While these factors are believed to play a role in the development of AGA in post-menopausal women, it's important to note that the exact mechanisms can vary among individuals. Genetics also play a significant role in determining who is more susceptible to AGA and how it manifests.

  1. Are there any published studies regarding menopausal scalp hair quantity and quality and female pattern hair loss in perimenopausal and menopausal women who are administered micronized progesterone?

#Yes, there is one study from Chaikittisilpa et al., 2022 ( Chaikittisilpa S, Rattanasirisin N, Panchaprateep R, et al. Prevalence of female pattern hair loss in postmenopausal women: a cross-sectional study. Menopause. 2022;29(4):415-420. Published 2022 Feb 14. doi:10.1097/GME.0000000000001927) describing the higher prevalence of FPHL in postmenopausal women.

There i salso a paper from Ablon and collaborators (Ablon G, Kogan S. A Randomized, Double-Blind, Placebo-Controlled Study of a Nutraceutical Supplement for Promoting Hair Growth in Perimenopausal, Menopausal, and Postmenopausal Women With Thinning Hair. J Drugs Dermatol. 2021;20(1):55-61. doi:10.36849/JDD.5701) which is a randomized, double-blind, placebo-controlled study was to assess the safety and efficacy of this oral supplement to promote hair growth in perimenopausal, menopausal, and postmenopausal women with self-perceived thinning.

As reported in the paper there is also a work from 018 Fabbrocini and collaborators [2018] which found that post-menopausal women with an-drogenetic alopecia (AGA) had lower levels of estrogen and higher levels of androgens, such as testosterone and dihydrotestosterone (DHT), compared to post-menopausal women without hair loss.

Reviewer 2 Report

Comments and Suggestions for Authors

In the manuscript entitled "The menopausal transition: is the HF “going in menopause”?", the authors discuss the impacts of the menopausal transition on the hair follicle lifecycle. The topic is interesting, however, there are several concerns which should be addressed before further processing of the manuscript.

- the manuscirpt would be more attractrive if the opinion descripthion is more clear.

- there are several paragraph consisting of only one sentence. The structure of the mauscript should be improved to focus the topics.

- The full name of HF should be provided in the Title.

- the full names of the abbreviations should be provided when appearing for the first time. Then abbreviations shoud be used in the text.

Author Response

REVIEWER 2

In the manuscript entitled "The menopausal transition: is the HF “going in menopause”?", the authors discuss the impacts of the menopausal transition on the hair follicle lifecycle. The topic is interesting, however, there are several concerns which should be addressed before further processing of the manuscript.

- the manuscirpt would be more attractrive if the opinion descripthion is more clear.

#Thanks for observation. We revised the manuscript accordingly and also addes more details about some practical ways to prevent hair loss during menopause.

We also added the following line on lines 23-32

“Menopause and hair loss often coincide in many women. During menopause, the body goes through significant hormonal changes, particularly a decrease in estrogen levels. This hormonal shift can lead to various symptoms, including hair loss or thinning. The exact mechanisms behind this are complex and can involve genetic predispositions, changes in the hair growth cycle, and sensitivity to androgens (male hormones).

Hair loss during menopause can manifest as diffuse thinning or a widening of the part. It's important to note that not all women experience significant hair loss during men-opause, and the severity can vary from person to person.

In this short opinion, we speculate about the existence of hair follicle (HF) ‘s menopause separate from the clinical menopause experienced by women.”

- there are several paragraph consisting of only one sentence. The structure of the mauscript should be improved to focus the topics.

#Thanks for observation. We revised the manuscript accordingly.

- The full name of HF should be provided in the Title.

#Done

- the full names of the abbreviations should be provided when appearing for the first time. Then abbreviations shoud be used in the text.

#Done

Round 2

Reviewer 1 Report

Comments and Suggestions for Authors

The authors have adequately addressed all the questions, and I have no further comments. The manuscript is ready for publication.

Reviewer 2 Report

Comments and Suggestions for Authors

The manuscript has been improved.